# Effect of Oxide Particles on Microstructure and Mechanical Properties of the 45 Carbon Structural Steel

**DOI:** 10.3390/ma13051232

**Published:** 2020-03-09

**Authors:** Bin Chen, Jianhang Feng, Yongzhen Zhai, Zhonghua Sun, Hongbo Liu, Yanfei Jiang, Jun He, Huifen Peng

**Affiliations:** 1School of Materials Science & Engineering, Hebei University of Technology, Tianjin 300130, China; 201731804037@stu.hebut.edu.cn (B.C.); fengjianhang@hebut.edu.cn (J.F.); 1987049@hebut.edu.cn (Y.J.); 2HBIS Group Technology Research Institute, HBIS Group, Shijiazhuang 052165, China; zhaiyongzhen@hbisco.com (Y.Z.); sunzhonghua01@hbisco.com (Z.S.); liuhongbo@hbisco.com (H.L.)

**Keywords:** 45 steel, oxide dispersion strengthening, oxide particles, heterogeneous nucleation

## Abstract

Striking difference in density between the oxide and the steel results in difficulty in preparing oxide dispersion strengthened steel with large size parts or materials. In this research, Al_2_O_3_ and TiO_2_ particles were initially milled with the 20 steel, and then the mixture was heated to a molten state to form a master alloy, which was used as a raw material for further preparation of the object steel. It was found that homogeneous distribution of the oxide particles was obtained in the mass production of the steel. Moreover, the obtained 45 carbon structural steel presents fine microstructures, together with improved mechanical properties, especially the impact ductility. This should be attributable to the transformation from the introduced micro-size oxide particles to the nano ones, which act as heterogeneous nucleants that play an important role in grain refinement and dispersion strengthening for the steel, during the remelting of the master alloy.

## 1. Introduction

It is known that the increase in strength of materials generally results in a drop in their ductility [1]. However, the steels with both high strength and good ductility are highly expected to meet the increasing demands in industry. Precipitation/dispersion strengthening is reported to be one of the most effective methods of simultaneously improving strength and ductility as well as other in-use properties [2,3,4]. In addition to playing a good strengthening effect, the secondary phases in the matrix also produce a good effect of grain refinement and then cause the increase of the corresponding ductility. Furthermore, this kind of strengthening has the advantage of relatively simple processing, and has thus become a powerful method of preparing high-strength steels in combination with good ductility [5].

The secondary phase particles may form in steels through in situ or ex situ ways. In situ formation of the secondary phase is caused by chemical reactions in steels. For example, the high strength low alloyed (HSLA) steels [5], twinning-induced plasticity (TWIP) ones [6] and quenching and partitioning (QP) ones [7] mainly make use of chemical reactions between microalloying elements like Nb, Ti and V and carbon/nitrogen to precipitate the carbide/carbonitride particles, which result in a good precipitation strengthening. However, these carbides/carbonitrides tend to nucleate at crystal defects like dislocations and grain boundaries. This kind of nucleation greatly limits size, number and distribution of the secondary phase particles; therefore, it is difficult to obtain an optimum strengthening effect on these steels. On the other hand, Jiang et al. [8] obtained a maraging steel with tensile strength of 2.2 GPa and elongation of 8.2% by means of homogeneous coherent precipitation of the Ni(Al,Fe) intermetallics from the martensite matrix. That should be attributable to the Ni(Al,Fe) intermetallics distributed in the martensite with very high density of 10^24^ /m^3^ and very fine particles of 2.7 nm in diameter. 

It is noteworthy that the in situ formed secondary phase are generally fine nano-particles, which are easy to peel off during abrasion and then lead to deterioration in performance of the materials. In recent years, the fast development in nuclear plants and weight lightening for structural parts makes the people have to pay more attention to the safety in the nuclear plants and the stiffness in the structural parts. Therefore, the oxide-dispersion strengthening (ODS) steels and high modulus steels are urgently needed. Among them, the secondary phases such as Y_2_O_3_ [9], WC [10], SiO_2_ [11], Al_2_O_3_ [12], NbC [13] and so on are usually introduced by powder metallurgy, through which the secondary phase particles can uniformly distribute in the matrix and the optimum performance can be obtained in the prepared materials. Furthermore, their performances can be continuously adjusted, according to specific demands, by modulating the type, size, number and so on of the secondary phases. However, the biggest disadvantage of the powder metallurgy is the preparation of large-size materials or parts at low costs. Furthermore, angularities formed during milling in the particles are easy to give rise to high local stresses, which induce crack initiation in materials and, eventually, their premature failure [14]. 

In 1960s and 1970s, Hasegawa et al. [15] sprayed some particles like oxides, sulphides and carbides into the molten metals during casting, and found that the obtained materials presented higher strength and better thermal stability than those without any introduced particles. Unfortunately, they can only prepare small ingots of 10 kg in weight. Other methods like the cored-wire feeding [16], non-conventional vortex liquid metal casting process [17] and ultrasonic caviation-based solidification [18] can be used to prepare large volume steels or parts strengthened by fine oxide particles, but their manufacturing cost is greatly increased. 

Through mixing some oxide particles of Al_2_O_3_ and TiO_2_ with steel powder and then melting at high temperature, we prepared a “master alloy” with a density close to the iron or steel. The master alloy was used as a raw material to produce the object steel, together with some other constituents like alloying elements. By controlling the particle size of the introduced oxides in the master alloy, we avoid inhomogeneous distribution of the oxide particles in mass production of steels. It not only simplifies the steelmaking process, but also is favorable to prepare the large-size materials or parts strengthened by uniformly dispersed oxides. In this study, we explored the 45 carbon structural steel as the research object and investigated the existence form and distribution of the introduced oxide particles in the steel. Its microstructures and mechanical properties were analyzed. 

## 2. Materials and Methods 

### 2.1. Preparation of Materials

Industrial alumina (γ-Al_2_O_3_ as the main constituent) with Al_2_O_3_ purity of 99%, titanium powder with rutile-TiO_2_ content of about 90%, silica powder with SiO_2_ purity of 99% and boron oxide with B_2_O_3_ purity of 98% were used as starting materials. They were weighed at a mass ratio of 40:40:10:10, mixed and then milled to the particle size of 0.1 μm to 10 μm. The milled oxides were mixed with the 20 steel (in GB standard) particles (about 2 mm in diameter) at a mass ratio of 1:9, and then heated to 1500 °C and held at this temperature for 2 h. After casting into molds, “master alloy” nuggets with oxide particles homogeneously distributed in the steel were obtained.

The master alloy nuggets and some pig iron were weighed according to the chemical compositions of the 45 steel in GB standard, smelted by the DDVIF-50-100-2.5 vacuum induction furnace (HBIS Group, Shijiazhuang, China), and then casted into a square ingot in width of 150 mm and in length of 300 mm. The ingot was heated at 1150 °C for 2 h and then hot-rolled into a plate of 40 mm in thickness and 120 mm in width in the temperature range of 1100 °C to 900 °C. The plate was cut into square bars of 40 mm in width and 200 mm in length. At last, the square bars were reheated to 1150 °C and hot-rolled into small square bars of 21 mm in width. The small square bars were used as the experimental materials in this study. Conventional 45 steel with a diameter of 20 mm was adopted for comparison; its chemical compositions are shown in Table 1.

The experimental materials were quenched into water after austenizing at 850 °C for 30 min, and then tempering at 520 °C to 620 °C for 2 h. At the end of tempering, the samples were taken from the furnace and then cooled in air to ambient temperature. 

### 2.2. Material Characterization

In order to analyze the effect of oxide particles on inclusions formed in the steel, non-metallic inclusions were evaluated based on ISO 4967-2013 standard [19]. The samples were cut across the maximum longitudinal section with an area of 200 mm^2^, they were mechanically ground and polished. Dection of various inclusions was conducted under metalloscope with magnification of 100. The oxide particles contained in the steel were electrolytically extracted in an aqueous solution of 1 wt.% NaCl, 2 wt.% citric acid and 3 wt.% FeSO_4_ at a current density of 0.03~0.1 A/cm^2^ under the pH value of 2 to 3. The extracted products were washed with deionized water. After drying, the extracted products were hand-milled for 30 min in a zirconia mortar. A magnet was used to eliminate the cementite particles in the extracted products. Then the residual extracted products were employed for further analysis.

Samples used for metallographic observation were mechanically ground, polished, and then etched with 4% nital solution. Samples for the measurement of prior austenite grains (PAGs) were etched at 70 °C for 2 min using a mixture of 5% hydrochloric acid and supersaturated picric acid. Samples for electron backscattered diffraction (EBSD) observation were electropolished in ethanol solution containing 7% perchlorate acid at a current density of 450 mA/cm^2^ for about 1 min. Then the samples were observed under FEI Scios DualBeam FE-SEM (Field Electron and Ion Company, Hillsboro, OR, USA), using secondary electronic signal imaging at 20 kV and a step of 0.15 μm. Samples for TEM observation were mechanically ground to 30 μm in thickness and then thinned with a Struers TenuPol-5 twin-jet machine (Struers, Westlake, OH, USA) in a mixture of 80% acetic acid and 15% chromic acid and 5% water. Then the samples, together with the residual extracted products being fished for by copper grids, were observed under Tecnai F30 field emission transmission electron microscope (Field Electron and Ion Company, Hillsboro, OR, USA). Chemical compositions of the secondary phase particles were measured by means of the attached EDX equipment (Field Electron and Ion Company, Hillsboro, OR, USA). X-ray diffraction (XRD) measurements were conducted on Rigaku Smartlab X-ray diffractometer (Rigaku, Tokyo, Japan) at a scanning rate of 1°/min to detect phase constituents of the extracted products.

Samples for tensile test were 10 mm in diameter and 50 mm in gauge length, and they were measured on a WDW-300 KN universal testing machine (Kesheng, Jinan, China) at a loading speed of 2 mm/min. Samples for impact ductility measurements were 10 mm in both width and height and 55 mm in length with V-type notch, then were tested on a Sansi PTM 2200 impact tester (Sansi, Shenzhen, China). The density of the samples was measured by Archimedes principle.

## 3. Results

### 3.1. Existing Form of the Oxide Particles 

Figure 1 shows the distribution of the oxide particles in the polished master alloy. It can be seen from Figure 1a that white particles (marked by white arrows) are the added oxides that can be proved by EDX results shown in Figure 1c. Those particles are not agglomerated and are almost spherical in morphology in spite of some difference in their size (Figure 1b). Moreover, some microcracks, marked by white arrows in Figure 1b, appear on its surface, and facilitate cleavage of the oxide particles during smelting of the steel. The dark black areas marked by symbol “□” are sulphide inclusions taken by the 20 steel (Figure 1d). In comparison with single milled ones, the oxide particles in the master alloy present the size less than 5 μm. These results indicate that the molten 20 steel fully penetrated interspaces among the oxide particles during heating at 1500 °C. Furthermore, some chemical reactions occurred between the molten steel and the oxide particles, that causes angularities formed during milling for the oxide particles to become rounded (Figure 1b), and is favorable to a firm bond between the steel matrix and the oxide particles. As the oxide contained about less than 10%, the obtained master alloy shows a high density of about 6.0 g/cm^3^, which is very close to that of steels. By using it as a starting material of preparing steels, it is easy to prevent inhomogeneous distribution of the oxide particles.

Figure 2a,b are SEM photos of the 45 carbon structural steel after oxide addition. Under a low magnification, the same as that of Figure 1a, almost no oxide particles were found in the sample (Figure 2a). After magnifying to 10,000 times, very fine oxide particles with most of their size less than 100 nm were observed in the steel matrix. Figure 2c indicates that their constituents are very close to those shown in Figure 1c for the introduced oxide particles. Based on the arc distribution of the fine oxide particles in Figure 2b, it is deduced that most of them present at around grain boundaries of the austenite. Their size and distribution observed in Figure 2b is almost the same as what we observed under TEM in Figure 2d. These results prove that most of the oxide particles were shattered into several smaller ones under thermal shock during remelting of the master alloy. The fine nano-sized particles have little buoyancy and are easy to stay every part of the steel uniformly, and they, acting as heterogeneous nucleants, could greatly refine the microstructures of the steel and improve its mechanical properties. This kind of indirect introduction of the nano oxide particles into the steel is much more convenient than that of the direct one reported by Padgornik et al. [20] and can be easily produced in large scale. 

Figure 3a is the XRD pattern of the residual extracted products for the oxide-added steel. The strongest XRD peaks marked by symbol “□” are in agreement with those of the rutile-TiO_2_ phase, and the stronger XRD peaks denoted by symbol “○” are consistent with those of the alpha-Al_2_O_3_ phase. These results prove that TiO_2_ and Al_2_O_3_ are main compounds in the extracted product. In spite of allotropy transformation in Al_2_O_3_ from γ- to α-phase, these oxides are the same as what we introduced into the master alloy even after remelting. Some weak peaks, marked by symbols “∆” and “☆”, correspond to those of the Fe_3_C and TiO compounds, respectively. The Fe_3_C is the residual cementite, which is difficult to be completely removed from the extracted product by magnet, and the TiO phase is probably one of the constituents in the titiania slag.

TiO_2_ and Al_2_O_3_ fine particles were generally used to induce the acicular ferrite in the weld metal in order to refine its microstructure under high heat input welding [21,22,23], however, the TiO_2_ was found to be unstable and transformed into a variety of complex deoxidation products between TiO_2_ and TiO. Furthermore, both the introduced TiO_2_ and Al_2_O_3_ particles cannot be used as the separate heterogeneous nucleants, and generally combined with some other inclusions such as MnS or MnO. These results suggest that great difference exists between the remelting for the master alloy in the present and the welding through oxide metallurgy.

Figure 3b shows TEM photos of the extracted products. Most of them are spherical particles with a diameter less than 50 nm, which is consistent with what was observed in Figure 2b,d. Furthermore, the electron diffraction patterns shown in Figure 3c,d prove that the extracted oxide particles should be attributable to Al_2_O_3_ and TiO_2_, respectively. Though they broke into nano-particles from the micro-ones during remelting of the master alloy, they remained stable and did not transform to any other phases that are different from those reported during welding. Nevertheless, nano oxide particles are highly desired in the steels, because they play an important role in grain refinement and dispersion strengthening.

### 3.2. Microstructures 

Figure 4 shows the distribution of PAG boundaries of the steels quenched at 850 °C for 30 min. The PAG size of the oxide-added steel is calculated to be about 11.2 μm in diameter, while that of the conventional one is about 31.8 μm in diameter. These results suggest that the oxide addition is effective in suppressing austenite grain growth and refining its grains. 

Figure 5 shows the microstructures of the samples quenched at 850 °C. Their microstructures are characterized as martensite where the equiaxed PAGs are subdivided into several packets (Figure 5a,b). The packet size is refined as the PAG size decreases. The average intercept length (IL_AV_) of the packets is measured to be about 9.1 μm for the oxide-added steel and is around 13.5 μm for the conventional one. These packets are further subdivided into fine blocks (Figure 5c,d). The boundaries of the blocks are defined as grain boundaries with a misorientation angle higher than 15°. The IL_AV_ of the blocks is measured to be 0.71 μm and 1.13 μm, respectively, corresponding to the oxide-added steel and the conventional one. These results indicate that oxide introduction results in finer and more homogeneous martensitic structures formed in the steel. 

### 3.3. Mechanical Properties

Figure 6a shows the stress-strain curves of the steels quenched at 850 °C and then tempered at different temperatures, where apparent yielding phenomena are observed for all curves. With an increase in the tempering temperature, both of the steels present gradual decrease in stress, concomitant with an increase in strain. Mechanical properties calculated according to those curves prove that the oxide-added steel exhibits both yield strength, Rel, and tensile strength, Rm, as well as percentage reduction of area, Z, higher than the conventional one at similar elongation, A (Figure 6b,c). Furthermore, the former shows higher impact energy, A_KV_, and hardness, as shown in Figure 6d. For example, the sample tempered at 620 °C presents yield strength of 773 MPa, tensile strength of 830 MPa and hardness of Hv 229, together with impact energy of 140 J, elongation of 19.6% and percentage reduction of area of 65%. Under similar plasticity, the strengths are at least 20% higher, the hardness at least 12.8% higher and the impact energy at least 97% higher than those of the conventional one. Generally speaking, the increase in strength results in decrease in both ductility and plasticity of materials. Conversely, the present steel exhibits high strength and hardness in combination with good ductility. 

Tempering is an additional treatment for the quenched steels and is a process where the quenched martensite gradually transforms into the stable microstructure, accompanied with the decrease in crystal defeats such as dislocations and grain boundaries between martensite packets, martensite blocks and martensite laths and precipitation of carbide particles. The presence of more hard oxide particles hinders the movement of dislocations and grain boundaries. Then it is deduced that the higher density of grain boundaries between martensite laths in the quenched martensite, as shown in Figure 5c, leads to higher density of crystal defects remaining in the tempered microstructure of the oxide-added 45 steel than those of the conventional one. On the other hand, tempering at high temperature results in a tempered sorbite, i.e., fine Fe_3_C particles distributing in the ferrite matrix. More nucleation sites are supplied for precipitation of the Fe_3_C carbides and a much finer tempered sorbite is obtained in the oxide-added 45 steel than those of the conventional one. That should be the reason why the oxide-added 45 steel exhibits high strength together with good ductility, as shown in Figure 6. 

## 4. Discussion

Figure 7 shows the distribution of non-metallic inclusions in the steels. It indicates that no apparent A-type sulfide inclusions, B-type alumina ones and C-type silicate ones were found in the whole field of view, and only D- and DS-type inclusions were observed in both steels. The number of the D-type oxide inclusions for the oxide-added steel is less than that of the conventional one, especially for those in size larger than 8 μm (Table 2). Moreover, there is almost no DS-type globular inclusions in the oxide-added steel. Contrary to traditional belief, introduction of the oxide particles not only does not increase the amount of the non-metallic inclusions, but also greatly decrease their number, especially for the bigger inclusions.

Densities were measured to be 7.7077 g/cm^3^ and 7.8284 g/cm^3^ for the oxide-added steel and the conventional one, respectively. At a condition of almost same compositions (Table 1), decrease in density should be attributable to the addition of oxide particles. It was known that the density of Al_2_O_3_ is about 3.7 g/cm^3^ and 4.2 g/cm^3^ for TiO_2_ [24]. Given that the average density of the introduced oxides is about 4.0 g/cm^3^, the content of the oxide particles in the present 45 steel is calculated to be about 3.1%, and their absorptivity is about 33.5%. It means that the unabsorbed oxide particles are discharged into slags. During floating, they interact with non-metallic inclusions present in the steel. The non-metallic inclusions may attach themselves to the floating oxide particles and become the big ones, which further increases their possibility of floating into slags. Accordingly, less non-metallic inclusions remain in the oxide-added steel. 

It is known that only oxide particles in a size smaller than 1 μm, especially the nano ones, can become the strengthening phases in metal materials [25,26], which is similar to carbides, carbonitrides and so on. However, the industrial oxide particles are generally in microscale, milling them into nano size needs a complicated processing. Furthermore, direct introduction of nano oxide particles into the steel makes it difficult to obtain a homogeneous distribution in compositions due to their agglomeration. Accordingly, the direct adoption of the micro oxide particles is cost-efficient and convenient in the oxide dispersion strengthening steels. 

Spraying the oxide particles with the average size about 10 μm into the molten steel stream, Hasegawa et al. [15] could decrease them into nano ones through the addition of the controlling elements such as Columbium. Variation in size of the oxide particles should be attributable to chemical reactions between the Columbium and the oxide particles. Though an apparent improved performance is obtained in the steels, its cost is greatly increased due to the addition of the expense controlling elements like Columbium. 

Showing high melting point, Al_2_O_3_ is difficult to be wetted by the molten iron even at a temperature of about 1550 °C [27]. It brings trouble in preparing the master alloy with the oxide particles homogeneously distributing in the steel matrix. On the other hand, oxides like TiO_2_, SiO_2_ and B_2_O_3_ show melting points lower than Al_2_O_3_. If they are mixed with the Al_2_O_3_, it is known that a mixture with a mass ratio of Al_2_O_3_:TiO_2_:SiO_2_ being 40:40:10 presents the melting point about 1800 °C, based on the phase diagram in the system Al_2_O_3_-TiO_2_-SiO_2_ shown in Figure 8 [24]. Further adding some B_2_O_3_ with much lower melting point into the above ternary mixture, the obtained quaternary mixture with the mass ratio of Al_2_O_3_:TiO_2_:SiO_2_:B_2_O_3_ being 40:40:10:10 will exhibit much lower melting point. Moreover, inclusions present in the starting materials of oxides will further lower its melting point. At this condition, the quaternary mixture is easily wettable in the molten steel during heating at 1500 °C. Then a homogeneous distribution of the oxide particles in the steel can be obtained in the master alloy without adding the expense controlling elements reported by Hasagawa et al. [15].

Considering that the SiO_2_ and B_2_O_3_ were not contained in the extracted product, as shown in Figure 1a, it is deduced that those two compounds mainly act as fluxes for lowering melting point of the oxide mixture and there probably form some glassy phase with high SiO_2_ and B_2_O_3_ contents. The glassy phase has a good flowability and easily surrounds every Al_2_O_3_ or TiO_2_ particle, then blunts the angularities formed during milling in the oxide particles. The glassy inclusions cannot be used as heterogeneous nucleants to refine microstructure of the steel, accordingly they are discharged into slags during melting of the steel. 

At last, this application of oxide particles in the 45 carbon structural steel is only our preliminary research on preparation of the oxide dispersion strengthening steels in mass production. Further research on high-carbon martensitic steel and high-strength invar alloy is currently conducted, and will be published in the near future. 

## 5. Conclusions

The Al_2_O_3_ and TiO_2_ micro-particles were initially milled with the 20 steel and melted to form the master alloy, which was used as raw material to prepare the object steel. Our results prove that it is an effective method of producing high-performance oxide dispersion strengthening steel in large scale. Though the oxide particles were introduced in micron scale, they were crushed into nano-sized ones, which act as heterogeneous nucleants that play an effective role in grain refinement and oxide dispersion strengthening for the steel, during remelting of the master alloy. Remaining similar plasticity, the oxide-added 45 steel showed strength at least 20% higher, the hardness at least 12.8% higher and the impact energy at least 97% higher than those of the conventional one after quenching at 850 °C and then tempering at 620 °C.

## Figures and Tables

**Figure 1 materials-13-01232-f001:**
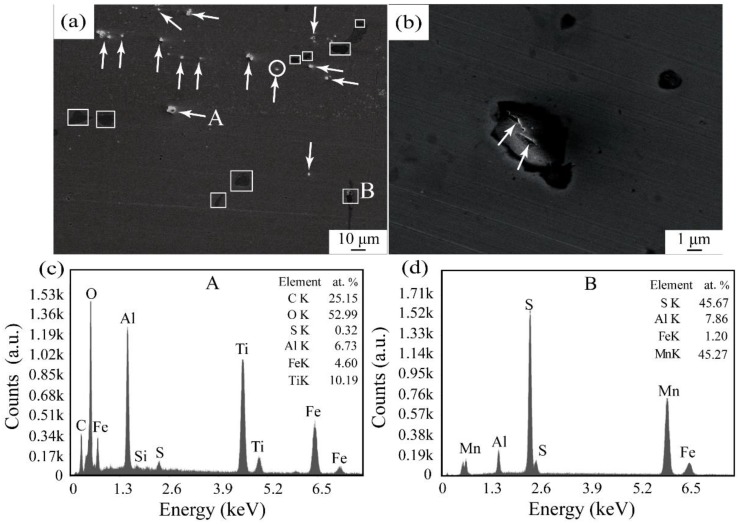
Distribution (**a**) of the oxide particles in the master alloy and EDX results of the points A and B in (**a**). (**b**) is the magnification to the circle in (**a**). (**c**), (**d**) are the EDX of the corresponding A, B particles, respectively.

**Figure 2 materials-13-01232-f002:**
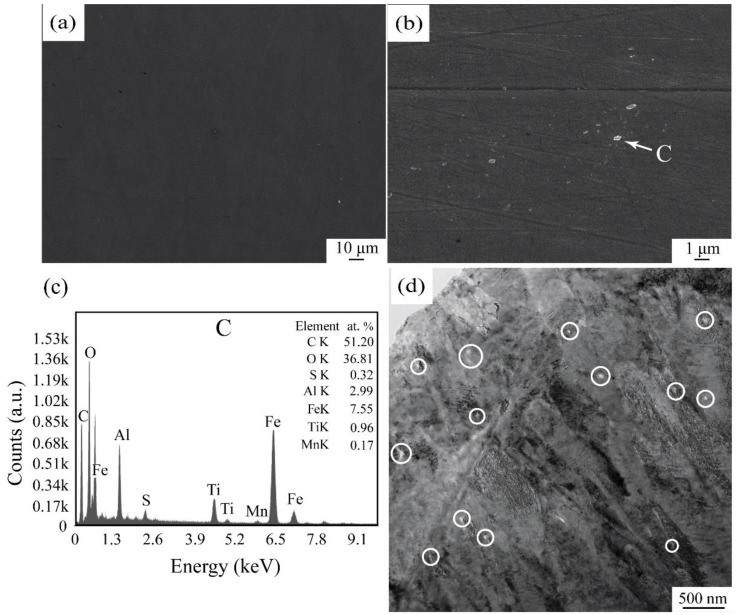
Distribution of the oxide particles in the 45 steel with oxide addition (**a**), (**b**) and (**d**). (**c**) is EDX result of the points C in (**b**).

**Figure 3 materials-13-01232-f003:**
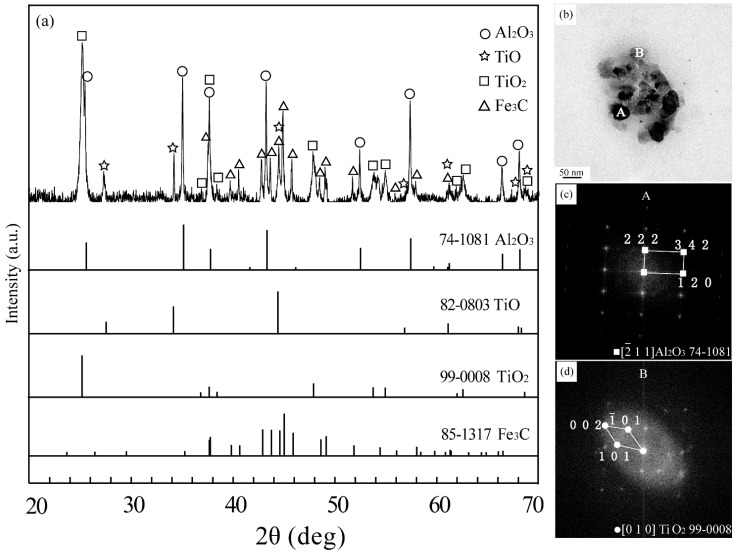
XRD spectrum (**a**) and TEM morphology (**b**) of the extracted particles. (**c**) and (**d**) are the electron diffraction patterns corresponding to the particles A and B in (**b**), respectively.

**Figure 4 materials-13-01232-f004:**
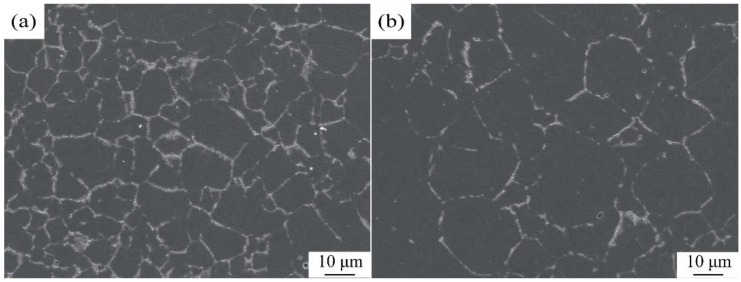
Distribution of PAG boundaries for the steels quenched at 850 °C, (**a**) oxide addition, (**b**) conventional.

**Figure 5 materials-13-01232-f005:**
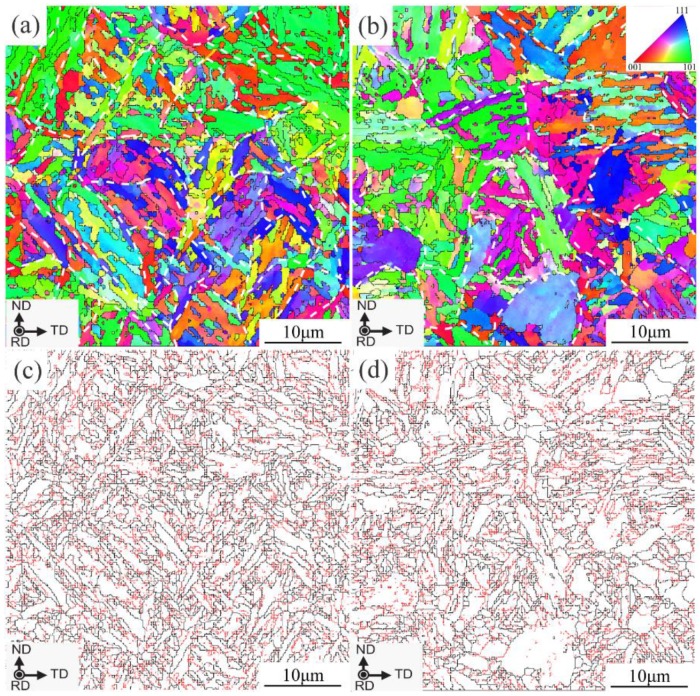
Microstructures of the quenched samples at 850 °C, (**a**,**c**) oxide addition, (**b**,**d**) conventional. The black and red lines in the image quality map (**c**,**d**) represents high angle boundaries of θ ≥ 15° and low ones of θ < 15°, respectively. RD means the rolling direction of the steel, TD the transverse direction and ND the normal direction.

**Figure 6 materials-13-01232-f006:**
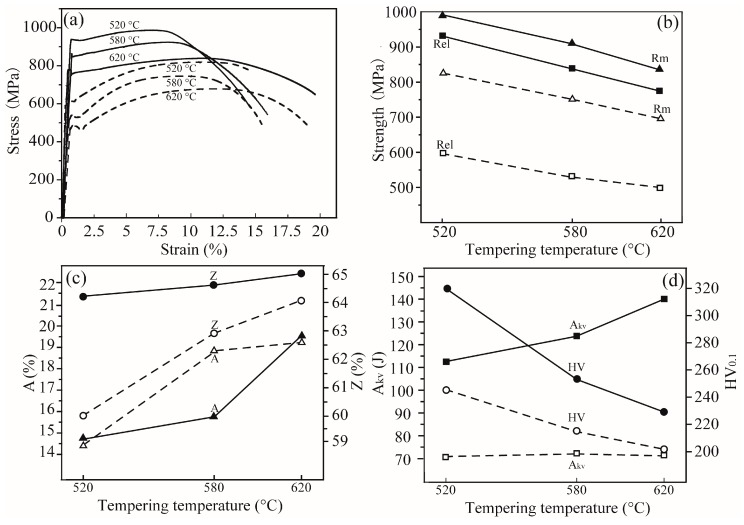
Stress-strain curves (**a**) of the steels after quenching and tempering at different temperatures and mechanical properties, (**b**) strength, (**c**) plasticity and (**d**) impact energy together with hardness. Solid lines and symbols mean the oxide-added steel, and dashed lines and open symbols denote the conventional one.

**Figure 7 materials-13-01232-f007:**
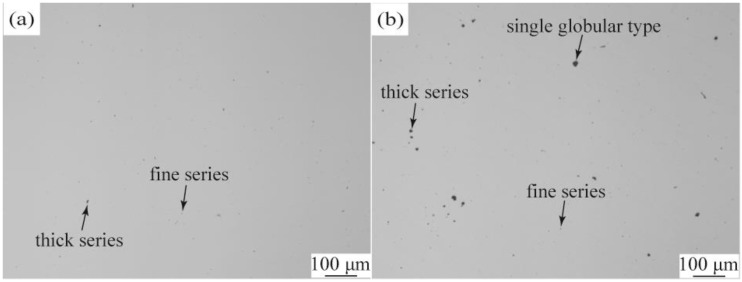
Inclusion distribution in the steels before (**b**) and after (**a**) oxide addition.

**Figure 8 materials-13-01232-f008:**
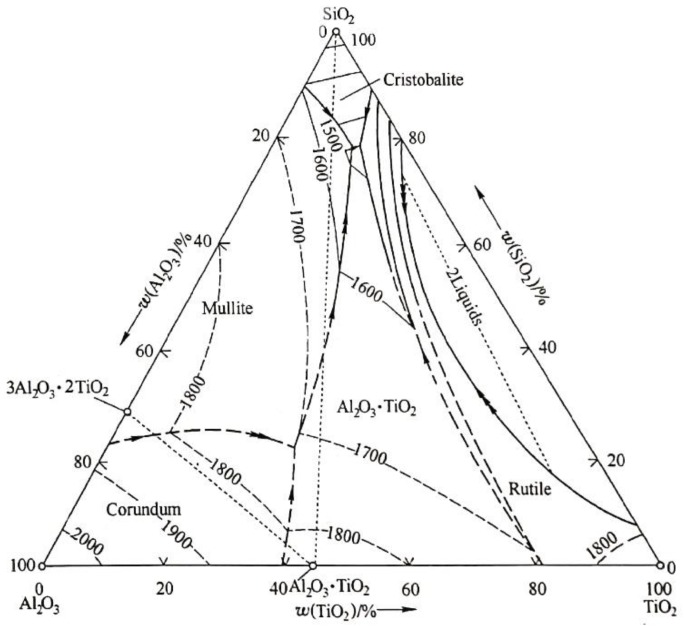
Phase diagram in the Al_2_O_3_-TiO_2_-SiO_2_ ternary system [24].

**Table 1 materials-13-01232-t001:** Chemical compositions of the steels used in this research (wt.%).

Oxide Addition	C	Mn	Si	P	S	Al	Ti	B	O
Yes	0.449	0.675	0.395	0.030	0.036	0.111	0.005	0.003	0.004
No	0.448	0.555	0.229	0.016	0.005	/	/	/	/

**Table 2 materials-13-01232-t002:** Evaluated inclusion grade in the steels based on Figure 7.

Oxide Addition	Group D (Globular Oxide Type)		Group DS (Single Globular Type) (13 μm~75 μm)
	Fine Series (3 μm~8 μm)	Thick Series (>8μm~13μm)	
Yes	2.13	0.5	No found
No	2.40	1.45	1.25

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
