# Peer review of "Effect of Oxide Particles on Microstructure and Mechanical Properties of the 45 Carbon Structural Steel"

_materials, 2020, doi:10.3390/ma13051232_

Round 1

Reviewer 1 Report

  1. Toughness = ductility
  2. The abbreviations used must be explained (HSLA, Twin, QP)
  3. p.2 line 4:... high modulus steels – what is it?
  4. To what environment was the material cooled from tempering temperature (p.3 in section 2.1)
  5. sec.2.1 – chemical signs of oxides are missing

- the weight percent of carbides is not clear in the context of the chemical composition; the whole chapter is very brief

6 Fig.1 missing axis descripion, percentage of Fe=? In spectral diagram

7.Fig.5: the standards refer to at least three samples for the tensile test

- Specify the stress in fig.5b  

  • what value is Ak in fig.5d? above the 30 HRC level, the hardness values ​​ should be in HV30

It is also necessary to pay attention to the economic profitability of this technological process in mass production, because steels with such and higher properties exist (microalloyed steels).

It is essential to achieve a homogeneous distribution of dispersed particles in the technology of manufacturing these steels. The way how was achieved it is kept silent in the article - it must to be explained

Reviewer 2 Report

The manuscript entitled “ Effect of oxide particles on microstructure and mechanical properties of the 45 carbon structural steel” was the subject of review. It concerns the preparation of ODS steel and studies its properties. The topics related to ODS steel are up to date and compelling. The manuscript presents interesting results and can be published. However, before acceptance the Authors, should …

  1. Explain the term ”secondary phase strengthening”
  2. Remove the references in Chinese
  3. Add the standard for indication of the 45 (or 20) steel
  4. Divide Fig. 1 and show it in magnification because
  5. the magnification used in Fig. 1a fails to state spherical shape of oxide particles
  6. the distribution of oxide particles is homogenous only in the top part of picture
  7. the Authors compare in Fig. 1 the master alloy and the 45 steel with oxide addition while in Fig. 3 the 45 steel with oxide addition with the conventional 45 steel. Why not show results for three of them?
  8. The quality of pictures (especially EDXs) is very poor. Moreover, the mark (e) looks almost the same as (c) in the printed version. It is hard to see the magnification bar.
  9. Explain all marks Rel, Rm, Z, A, Ak
  10. Present the discussion in Par. 3.3 more clearly. For example, there is a lack of any information in the text about studied samples. What is compared: the 45 steel with oxides and master alloy or the conventional one.
  11. Consider the role of structural defects like dislocations and their impact on the studied properties in Par. 3.3. Can we say something about it?
  12. Explain what is compared in Fig. 6. The 20 steel and “master alloy”?
  13. Add the references to prove the indentation starting from the sentence “It is known that only oxide particles in size less than 1 μm, especially the nano ones, can become strengthening phases in metal materials, similar to carbides, carbonitrides and so on.”

Round 2

Reviewer 1 Report

Editing the article has helped, is more comprehensible and accurate in the expression of ideas

Fig. 6: axis title: stress (MPa), strain (%)   
